# An ecological study of the spatiotemporal dynamics and drivers of domestically acquired campylobacteriosis in Ireland, 2011–2018

**Martin Boudou**[1]*, **Coilín ÓhAiseadha**[2], **Patricia Garvey**[3], **Jean O'Dwyer**[4,5], **Paul Hynds**[1,5]*

**1** Environmental Sustainability & Health Institute (ESHI), Dublin, Republic of Ireland, **2** Department of Public Health, Health Service Executive (HSE), Dr. Steevens' Hospital, Dublin, Republic of Ireland, **3** Health Protection Surveillance Centre, Dublin, Republic of Ireland, **4** School of Biological, Earth and Environmental Sciences, Environmental Research Institute (ERI), University College Cork, Cork, Republic of Ireland, **5** Irish Centre for Research in Applied Geosciences (iCRAG), University College Dublin, Dublin, Republic of Ireland

\* hyndsp@tcd.ie (PH); Martin.boudou@gmail.com (MB)

**Data Availability Statement:** Due to ethical considerations (sensible information due to GDPR regulations and associated data sensitivity at

## Abstract

In 2021, Campylobacteriosis was the main gastrointestinal disease in the European Union since 2007 according to the European Centre for Disease Prevention and Control. In the Republic of Ireland, the incidence of the disease is particularly high with approximately 3,000 cases per annum, raising significant concerns for national health authorities with an expected increase in the number of cases in the light of climate change. The current study sought to assess the spatio-temporal patterns of campylobacteriosis in the Republic of Ireland using 20,391 cases from January 2011 to December 2018. An ensemble of spatial statistics techniques including seasonal decomposition, spatial clustering and space-time scanning, were used to elucidate the main individual and spatio-temporal characteristics of the disease in the country. Findings revealed that cases from the paediatric age group (i.e., under 5 years old) were more likely to occur in rural areas (aOR: 1.1.27, CI 95% 1.14–1.41) while cases from the intermediate age group (i.e., >5 & <65 years old) were associated with urban living (aOR: 1.30, CI 95% 1.21–1.4). The disease exhibited a peak during Irish summer, with a stronger seasonal signal reported in counties located on the Western part of the country. Infection hotspots were more likely to occur in urban areas, and more particularly on the Southern part of the island and around the main metropolitan areas. Overall, research findings pointed out the influence of local and spatio-temporally specific socio-demographic and environmental risk factors (i.e., cooking habits, local weather, dietary types) therefore highlighting the need for initiating spatio-temporally targeted health management and surveillance strategies.

individual and/or Small Area level), data are not publicly accessible. Infection data used for the current study are however available upon request from the CIDR Committee and Health Protection Surveillance Centre website (https://www.hpsc.ie).

**Funding:** -Initial of the funded author: PH -Grant number: 2019-CCRP-MS.62 -Funder name: Environmental Protection Agency Ireland - https://www.epa.ie/ The funders had no role in study design, data collection and analysis, decision to publish, or preparation of the manuscript.

**Competing interests:** The authors have declared that no competing interests exist.

## 1. Introduction

Campylobacteriosis is caused by zoonotic pathogenic *Campylobacter* species, and has consistently been identified as the most frequently diagnosed cause of foodborne bacterial gastrointestinal infection in the European Union since 2007 [1, 2]. Campylobacteriosis symptoms typically include abdominal pain, nausea, diarrhoea, vomiting, fever and dehydration [3], however, in the most severe cases, the infection can trigger the onset of sequelae such as Irritable Bowel Syndrome (IBS), reactive arthritis or Guillain-Barré Syndrome (GBS), a peripheral neurological disease potentially leading to muscle paralysis, respiratory problems and death [4, 5]. Within addition to the clinical consequences of campylobacteriosis, the infection is also responsible for substantial economic costs linked with healthcare and productivity losses. Tam and O'Brien [6] previously estimated an average cost of £85 per acute case of campylobacteriosis in the United-Kingdom, and therefore approximately 250–300% higher than equivalent costs per case of norovirus and rotavirus (i.e., £30). While latest Figures show a relatively stable incidence rate over the past decade across the European Union, the prevalence of both endemic and sporadic campylobacteriosis is expected to increase significantly in light of global warming [2]. For example, Kuhn et al. [7] recently estimated that the number of campylobacteriosis cases is expected to increase by almost 200% in Northern-Europe by the end of the century due to increasing annual temperatures and an increased frequency and magnitude of heavy rainfall events and heatwaves.

*Campylobacter* spp. is the most frequently diagnosed cause of bacterial gastroenteritis in the Republic of Ireland (ROI) with approximately 3,000 cases per annum and a national crude incidence rate of 62.9 cases/100,000 during 2021, therefore much higher the EU mean (44.5 cases/100,000) [2]. To date, a few studies have been conducted on the mechanistic epidemiology of human campylobacteriosis in the ROI. Danis et al. [8] reported findings from a matched case-control study of 196 cases of infection and identified consumption of chicken, lettuce and food from takeaways/restaurants as primary risk factors for acquisition of campylobacteriosis in the ROI. O'Connor et al. [9] described notified cases of campylobacteriosis (2004–2016) by age, gender, geographical area, and season, with incidence rates higher in, males (aIRR 1.15, 95% confidence intervals (CI) 1.12–1.19), those aged <5 years compared with the lowest incidence age group (45–64 years) (aIRR 4.65, 95% CI 4.43–4.88), other seasons compared with winter and all other areas compared with the north-east area (aIRR range 1.22–1.71, p-values < .001). While both studies present findings pertaining to the likely risk factors for contracting campylobacteriosis in the ROI, to date, no comprehensive study has been undertaken to examine the spatio-temporal characteristics of infection at a national scale, thus effectively minimising any potential spatial or temporal biases. Spatial statistics and time-series analysis have previously been employed to identify the primary spatio-temporal drivers of campylobacteriosis. For example, Green et al. (2020) employed spatial scan statistics to identify high and low rate clusters of campylobacteriosis from 1996 to 2004 in Manitoba, Canada [10]. Similarly, Marek et al. (2015), used spatio-temporal Kriging spatial interpolation and spatial scan statistics to describe the spatio-temporal patterns of campylobacteriosis in Czech Republic (2008–2012) [11].

Accordingly, the current study sought to investigate and present a comprehensive analyses of the primary spatio-temporal characteristics of domestically-acquired campylobacteriosis from 2011 to 2018 using a series of machine learning (i.e., seasonal decomposition, clustering) and spatial statistics (i.e., spatial autocorrelation, space-time scanning) techniques.

## 2. Methods

### 2.1 Infection data

On the 31st of March 2022, irreversibly anonymised laboratory-confirmed cases of campylobacteriosis occurring between the 1st of January 2011 and the 31st of December 2018 were acquired from the Computerised Infection Database Reporting (CIDR), the national database of notifiable infectious disease in the ROI (http://www.hpsc.ie/CIDR). Individual-level data linked to specific cases included age, gender, country and county of infection. Cases associated with recent international travel (N = 81) and secondary cases associated with outbreaks (N = 159) were excluded from the current study. Thus, 20,391 primary endemic cases of campylobacteriosis were included for geo-linkage. All individual cases of laboratory-confirmed infection were geographically linked to one of the 2016 CSO Census Small Areas (SAs), the smallest administrative unit existing in the ROI (N = 18,641), via the geocoding protocol developed by Domegan et al. [12].

Overall, 14,338 of endemic campylobacteriosis cases (70.6% of total cases) were successfully spatially joined to one Small Area. Contrastingly, 6,053 (29.4% of total cases) couldn't be linked to a single Small Area, due to lack of address clarification during the collection protocol. The primary attributed settlement type for each SA was derived from the 2016 CSO Census Classification (CSO, 2019), as follow (Fig 1):

- Urban areas, main city centres and satellites towns (N = 10,771, 57.8% of total SAs),

- Commuter areas, areas with a moderate or high urban influence (N = 4,769, 25.6% of total SAs),

- Rural areas, remote and/or rural areas (N = 3,101, 16.6% of total SAs).

Fisher exact tests were employed to identify associations between case gender, age and settlement types using R and R Studio (v 4.0.5).

### 2.2 Seasonal decomposition and temporal clustering

Seasonal decomposition was undertaken using the Seasonal and Trend (STL) decomposition via LOESS approach (Locally Estimated Scatterplot Smoothing). The STL method decomposes a time-series into three individual components (seasonal variation, general trend and residuals) whereby the total sum is equal to the monthly case number [13]. Seasonal decomposition was performed through the use of an additive model and based on the monthly mean incidence rate of infection per 100,000 people (N = 20,391) to identify the primary temporal signals of campylobacteriosis across the country. Subsequently, a series of decompositions were undertaken for each county in the ROI (N = 26, Fig 1), using monthly mean and county-specific incidence rates of infection per 100,000 people. A similar clustering approach was used by Djennad et al [14] to identify identical seasonal patterns of campylobacteriosis between thirty-three sub-regions of England and Wales. For the current study, seasonal signals obtained from individual (i.e., county level) seasonal decompositions were employed as inputs for cluster analysis to identify counties exhibiting similar temporal patterns, using a hierarchical clustering approach via the Ward2 algorithm. The Ward2 method minimises the clustering criterion initially produced by the Ward method [15]. Tukey Pairwise Comparison and Fisher exacts tests were employed to identify potential differences and similarities (i.e., age, gender, settlement types) between identified clusters. All analyses were performed on R & RStudio suite (v 4.0.5), using the base STL() function and the "hclust" function from the "stats" package. Irish seasonal calendar, traditionally used in the ROI and different from other European countries,

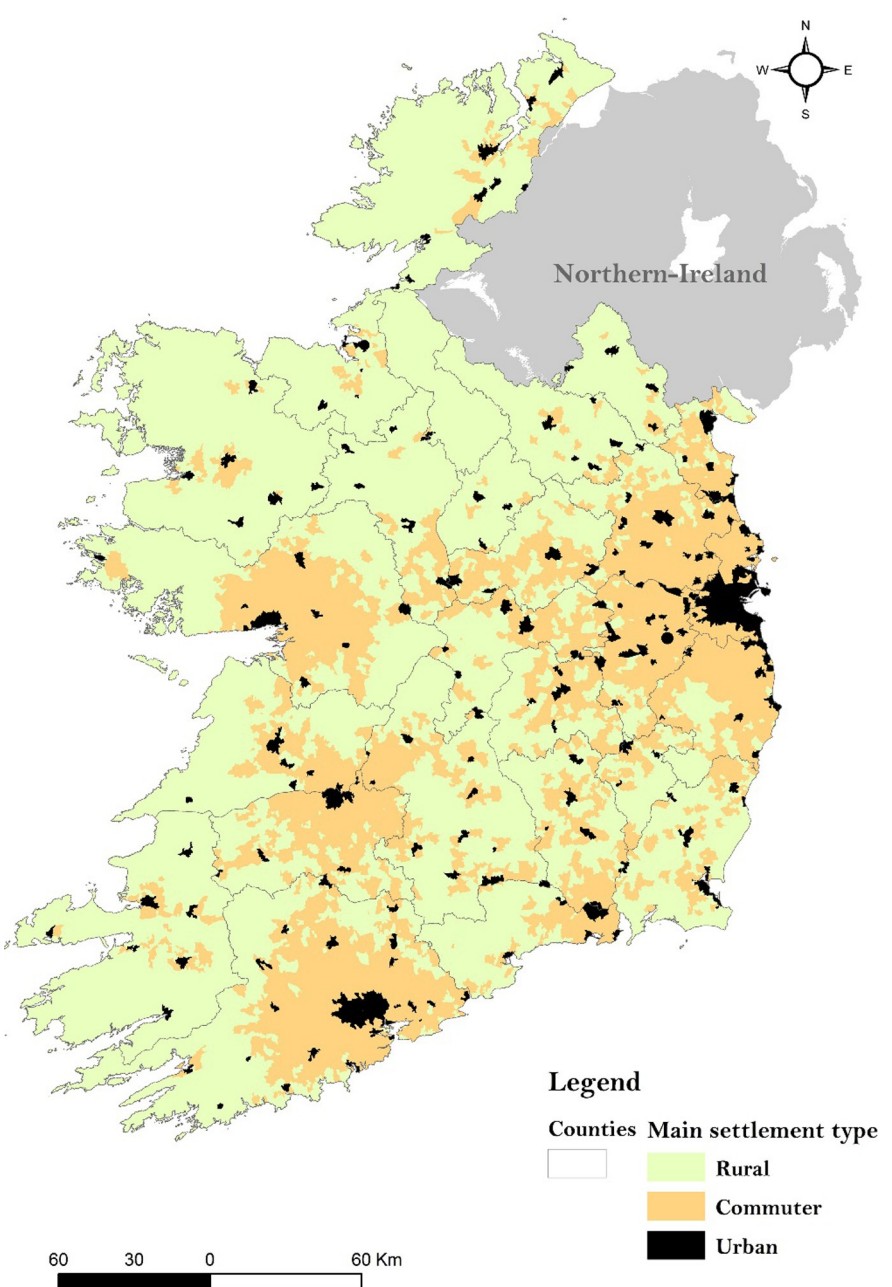

**Fig 1. Spatial distribution of settlement types (rural, commuter and urban areas) delineated by small area across the Republic of Ireland - derived from the 2016 CSO classification.**

was employed for describing campylobacteriosis seasonal patterns (i.e., Winter: November to January, Spring: February to April, Summer: May to July, Autumn: August to October).

## 2.3 Spatial autocorrelation

Hot-spot analyses were undertaken using the Getis-Ord Gi* geostatistical tool in ArcGIS (v 10.7), with all geographically referenced cases (N = 14,338). The Getis-Ord GI* statistic, initially developed by Getis and Ord [16] permits identification of spatial clusters characterised

by significantly low or high levels of association. A simplified version of the Gi* statistical equation may be presented as follows:

$$G_{i*} = \frac{\sum_{j=1}^{n} W_{ij}X_j}{\sum_{j=1}^{n} X_j}$$

where, Gi represents the spatial dependency of incident (*i*) over all events (n), $X_j$ is the magnitude of variable *X* at the incident location (*j*) over all events (*n*), and $W_{ij}$ is the spatial relationship (weighting value) between events (*i* & *j*) [17]. In the current study, crude incidence rate per 100k people for each SA was used to identify hot/cold spots of campylobacteriosis infections across Ireland. Fisher exact tests were consecutively conducted between the presence of identified hot/cold spots and attributed settlement type.

## 2.4 Space-time scanning and space-time cluster frequency mapping

A space-time scanning method was applied to geo-referenced cases of campylobacteriosis (N = 14,338) using SaTScan v.10 (Kulldorf and Information Management Services, Inc., MA, USA). A retrospective space-time scanning method was employed for each year of the infection dataset (2011 to 2018), with a discrete Poisson model selected to account for the high spatial resolution of SAs (N = 18,641). The population in each SA reported during the last national Irish census in 2016 was used as the underlying control parameter. Other parameters included definition of a 10-case minimum threshold per cluster to avoid identification of multiple small clusters within the same SA during the same timeframe. Similarly, a maximum radius of 50 kilometres per cluster was selected to account for the low number of cases per SA (i.e., maximum number of cases per SA was 14 during the study period). Lastly, a monthly data aggregation was used, with a maximum cluster duration of 3 months to account for existing seasonal patterns of campylobacteriosis in the ROI. Similar SaTScan parameterisation has previously been successfully employed by Boudou et al. [18] and Cleary et al. [19] for cryptosporidiosis and verotoxigenic *E. Coli* enteritis in Ireland, respectively.

Based on annual space-time scans, a space-time cluster recurrence index was developed, by spatially overlaying the presence/absence (1/0) of annual space-time clusters across all Small Areas across the country [18–20]. Accordingly, a cluster recurrence is calculated for each SA, ranging from 0 (i.e., no cluster detection over the entire time-series) to 8 (i.e., space-time cluster identified at least once per year over the entire time-series). The final index and mapped dataset provides a temporally-adjusted hot/cold-spot map of infection risk. Fisher Tests were performed to detect l associations between space-time cluster presence, elevated space-time cluster frequency (i.e., ≥2, equivalent to the 3rd quartile of the IQR), and settlement type.

## 3. Results

### 3.1 Incidence and characteristics of campylobacteriosis in the Republic of Ireland, 2011–2018

Laboratory-confirmed cases of campylobacteriosis exhibited a higher overall incidence among the paediatric (≤ 5 years old) sub-population and males, with 4,627 cases (22.7% of total) and 11,121 (54.5% of total) cases, respectively (Table 1 and Fig 2). Paediatric cases were statistically more likely to be male (aOR: 1.27, CI 95% 1.14–1.41) while the intermediate (aOR: 1.12, CI 95% 1.06–1.19) and older age groups (aOR: 1.14, CI 95% 1.05–1.23) were more likely to be female.

**Table 1. Gender, age group and results of fishers exact tests for campylobacteriosis in the Republic of Ireland (2011–2018) (N = 20,391).**

| Gender | Cases | CI 2.5% | Adj. OR | CI 97.5% | p-value |
|---|---|---|---|---|---|
| ≤ 5 y. | | | | | |
| F | 1,877 | 0.73 | 0.78 | 0.83 | <0.001 |
| M | 2,731 | 1.18 | 1.27 | 1.35 | <0.001 |
| > 5 & < 65 y. | | | | | |
| F | 5,876 | 1.06 | 1.12 | 1.19 | <0.001 |
| M | 6,781 | 0.85 | 0.90 | 0.95 | <0.001 |
| ≥ 65 y. | | | | | |
| F | 1,746 | 1.05 | 1.14 | 1.23 | 0.001 |
| M | 2,006 | 0.82 | 0.89 | 0.96 | 0.002 |

A significantly higher incidence of campylobacteriosis occurs in urban areas, equating to approximately 66.6% (N = 9,544) of all geo-linked cases (N = 14,338) (Table 2). Similarly, a significant positive association was found between female cases and urban settlements (aOR: 1.09, CI 95% 1.01–1.17). Male cases were more likely to occur within commuter regions (aOR: 1.17, CI 95% 1.07–1.29) and rural areas (aOR: 1.005, CI 95% 0.92–1.09). A positive association was identified between paediatric cases and both rural (aOR: 1.259, CI 95% 1.14–1.39) and commuter areas (aOR: 1.27, CI 95% 1.14–1.41) while cases within the intermediate (> 5 & < 65 years old) and elderly (≥ 65 years old) age groups were positively associated with urban (aOR: 1.3, CI 95% 1.21–1.40) and rural living (aOR: 1.250, CI 95% 1.11–1.40), respectively.

Fig 3 shows the spatial distribution of the annual incidence rate of campylobacteriosis across counties from the ROI (N = 26). From 2011 to 2018, a mean crude incidence rate (CIR) of 55 cases/100,000 was notified nationally. The highest CIR was reported for county Waterford in the South of the country with 77.2 cases/100,000 annually, followed by counties Westmeath (midlands, 75.7 cases/100,000) and Kilkenny (south, 74.2 cases/100,000). Conversely,

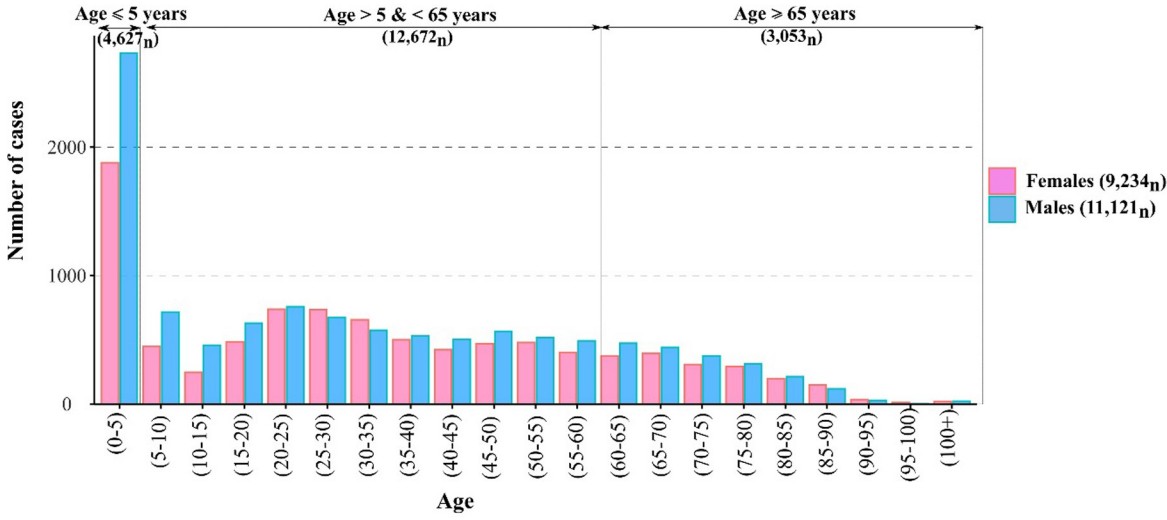

**Fig 2. Age and gender distributions of campylobacteriosis cases in the Republic of Ireland (2011–2018)–N = 20,391.**

**Table 2. Gender and age range distribution per settlement types - results from the fisher tests for campylobacteriosis infections in the Republic of Ireland (2011–2018)–geo-linked dataset (N = 14,338)–NS: Non significant p-value.**

| Sub-population | Cases | CI 5% | Adj. OR | CI 95% | p-value |
|---|---|---|---|---|---|
| Commuter (N = 2,055 (14.3%)) | | | | | |
| F | 878 | 0.78 | 0.858 | 0.94 | 0.002 |
| M | 1,177 | 1.07 | 1.173 | 1.29 | 0.001 |
| ≤ 5 y. | 545 | 1.14 | 1.267 | 1.41 | <0.001 |
| > 5 & < 65 y. | 1,245 | 0.82 | 0.905 | 1.00 | 0.043 |
| ≥ 65 y. | 265 | 0.73 | 0.846 | 0.97 | 0.017 |
| Rural (N = 2,739 (19.1%)) | | | | | |
| F | 1,259 | 0.92 | 1.001 | 1.09 | 0.002 |
| M | 1,480 | 0.92 | 1.005 | 1.09 | 0.001 |
| ≤ 5 y. | 716 | 1.14 | 1.259 | 1.39 | <0.001 |
| > 5 & < 65 y. | 1,554 | 0.68 | 0.742 | 0.81 | 0.044 |
| ≥ 65 y. | 464 | 1.11 | 1.250 | 1.40 | 0.019 |
| Urban (N = 9,544 (66.6%)) | | | | | |
| F | 4,452 | 1.01 | 1.087 | 1.17 | 0.019 |
| M | 5,092 | 0.85 | 0.913 | 0.98 | 0.011 |
| ≤ 5 y. | 2,002 | 0.69 | 0.744 | 0.81 | <0.001 |
| > 5 & < 65 y. | 6,163 | 1.21 | 1.302 | 1.40 | <0.001 |
| ≥ 65 y. | 1,383 | 0.84 | 0.930 | 1.03 | 0.146[NS] |

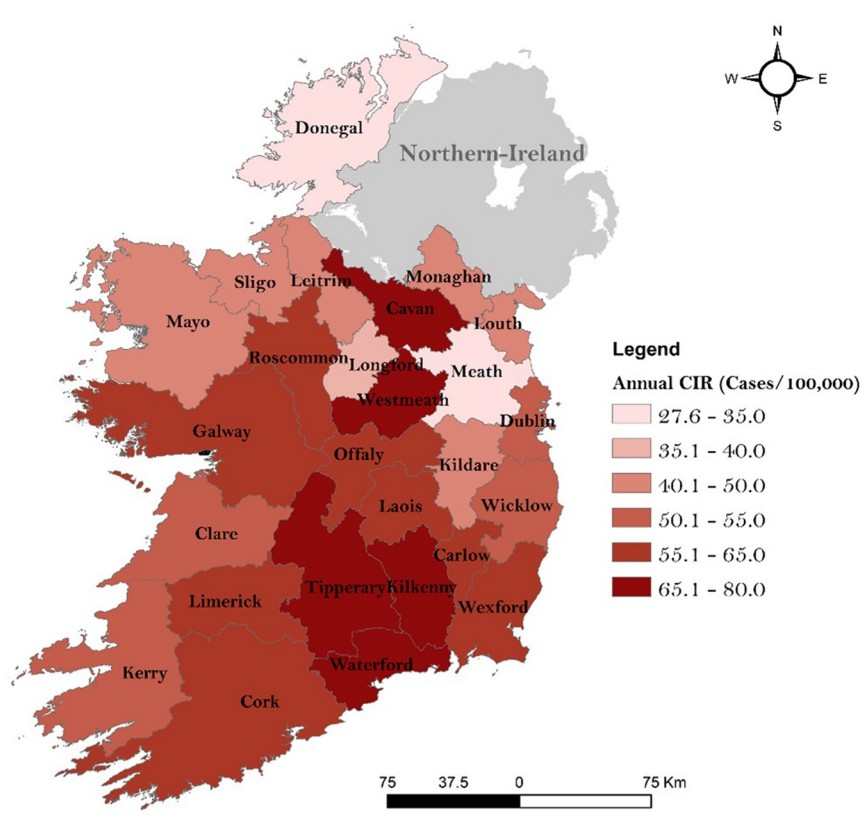

**Fig 3. Annual crude incidence rates of campylobacteriosis (cases/100,000 inhabitants) per county, 2011–2018.**

lowest annual CIRs were notified for counties Meath (27.6 cases/100,000) and Donegal (34.1 cases/100,000).

## 3.2 Temporal patterns of campylobacteriosis

As shown (Fig 4), highest incidence rates occur during Irish summer (May to July, N = 7,155) peaking in May (N = 2,664), while lowest incidence rates occur during Irish Winter (N = 3,911), from November (N = 1,524) to January (N = 1,258). Seasonal signals obtained from seasonal decomposition (Fig 5), confirms the highest incidence rate of campylobacteriosis during Irish summer, peaking in May (2.54 cases/100,00). The long-term trend signal (Fig 5) exhibited a general increase over the study period, with an increase of approximately 0.16 cases/100,000 year on year from 2011 to 2018. The lowest annual CIR occurred during 2013 (48.11 /100,000), with 2013 also exhibiting a marked decrease in the long-term trend signal (Fig 4), while highest annual incidence was reported in 2018 (61.96 cases/100,000), peaking in May 2018 (8.63 cases/100,000). The highest number of excess cases (i.e., outlying events) was reported during May 2014 (+ 129 cases/100,000 above average monthly mean), January 2016 (+1.12 cases/100,000) and March 2017 (+1.01 cases/100,00). Similarly, a higher incidence of cases was identified month on month from March to June 2018, while marked negative residuals were identified during May 2012 and May 2015 (- 1.12 cases/100,000).

## 3.3 Seasonal clustering

Three primary clusters characterised by similar seasonal patterns were identified (Fig 6A and Table 3). The first cluster (Cluster 1) includes counties located in the North and East of the ROI and comprised 10 counties with a total population of approximately 1.6 million people. The second cluster (Cluster 2) was associated with a total of 14 counties located on the Western part of the ROI, with a total population of approximately 2.6 million people. Lastly, Cluster 3 was comprised of just 2 counties, namely Westmeath and Kilkenny, having a total population of approximately 180,000 people.

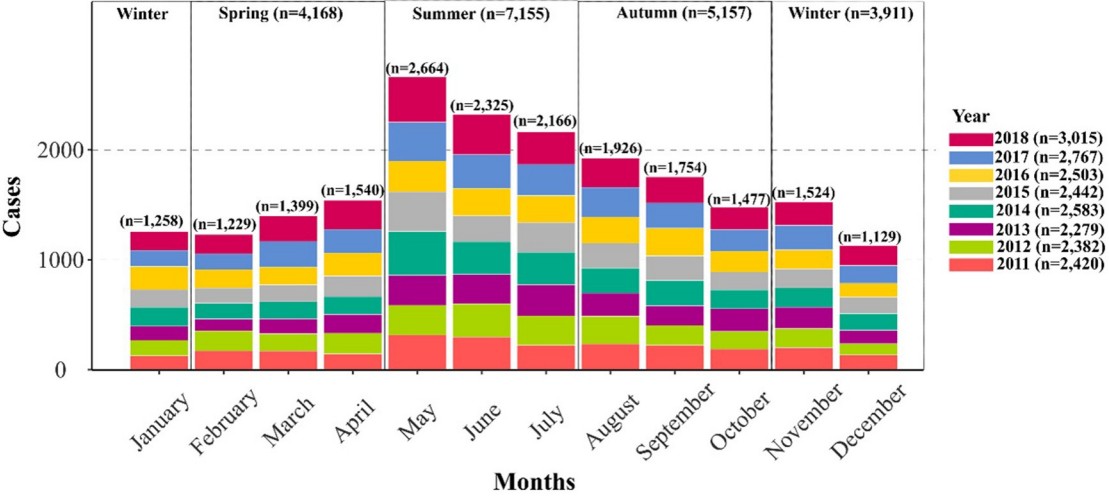

**Fig 4. Monthly, seasonally and annual occurrence of campylobacteriosis in the Republic of Ireland, 2011–2018 (N = 20,391).**

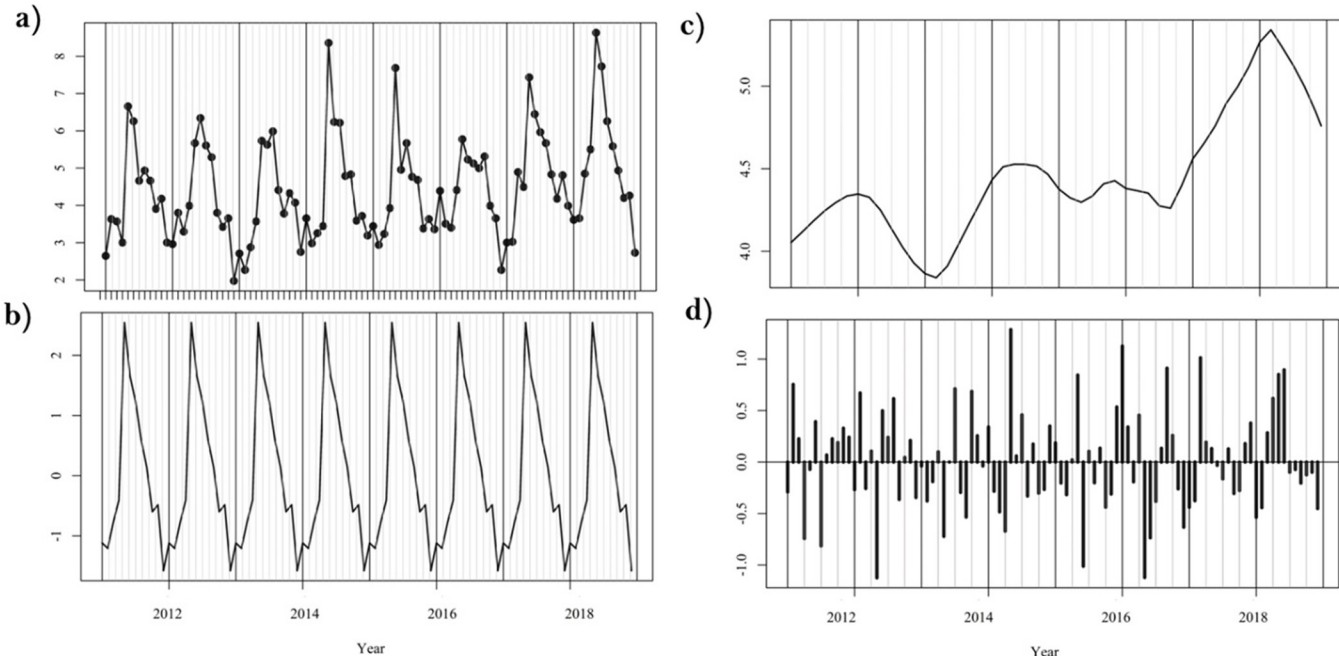

**Fig 5. Seasonal decomposition of campylobacteriosis infection in the Republic of Ireland (2011–2018)–overall dataset (N = 20,391).** a) Raw data, b) Seasonal variations, c) General trend, d) Residuals.

Fig 6B presents the seasonal variations obtained for each county clusters. The highest incidence of cases was reported within the Cluster 3 with an annual mean incidence of 78.06 cases/100,000 (monthly incidence of 6.02 cases/100,000), followed by Cluster 1 (yearly: 68.53, monthly: 4.92 cases/100,000) and Cluster 2 (yearly: 46.54, monthly: 4.09 cases/100,000). All three clusters observed a main infection peak in May, with a stronger signal obtained for Cluster 3 with an average of 12.5 cases/100,000 people during this month. Similarly, the lowest incidence was reported during December for all three clusters, ranging from 2.4 cases/100,000 (Cluster 2) to 3.2 cases/100,000 (Cluster 1). A secondary peak of infection was noticeable during November in Cluster 3 (6 cases/100,000). Conversely, Cluster 2 exhibited a more regular pattern of seasonal variations in comparison with Cluster 1 & 3, with less monthly deviations observed. Lastly, a higher incidence of cases was noticeable in Cluster 1 compared to Cluster 2.

The Tukey Pairwise Comparison test (S1 Table) revealed that mean case age within Cluster 2 (34.7 years) was significantly higher than the mean case ages observed in Clusters 1 and 3 (+3.9 and +4.2 years, respectively). A significantly larger proportion of categorically urban cases occurred within Cluster 2 (57.1% of total cases, n = 5,856), compared with Cluster 1 (37.6%, n = 3,370) and Cluster 3 (16.9%, n = 318). As such, urban cases were approximately 3 times more likely within Cluster 2 (OR: 3.03, S2 Table). Conversely, significant positive associations were identified between both Clusters 1 and 3 and rural/commuter areas.

### 3.4 Spatial autocorrelation–GetIs-OrdGi

Fig 7 shows the spatial distribution of identified static hot and cold-spots of campylobacteriosis in the ROI from 2011 to 2018, obtained via GetIs-OrdGi. Overall, 27.1% of total SAs were categorised as infection hotspots (n = 5,060) and 15.4% as cold-spots (n = 2,863) (S3 Table). County Kilkenny reports the highest percentage of SAs being identified as hotspots (68.6% of total SAs), followed by Waterford (67.2%) andCork (61.1%) . Notably, hotspots of infection

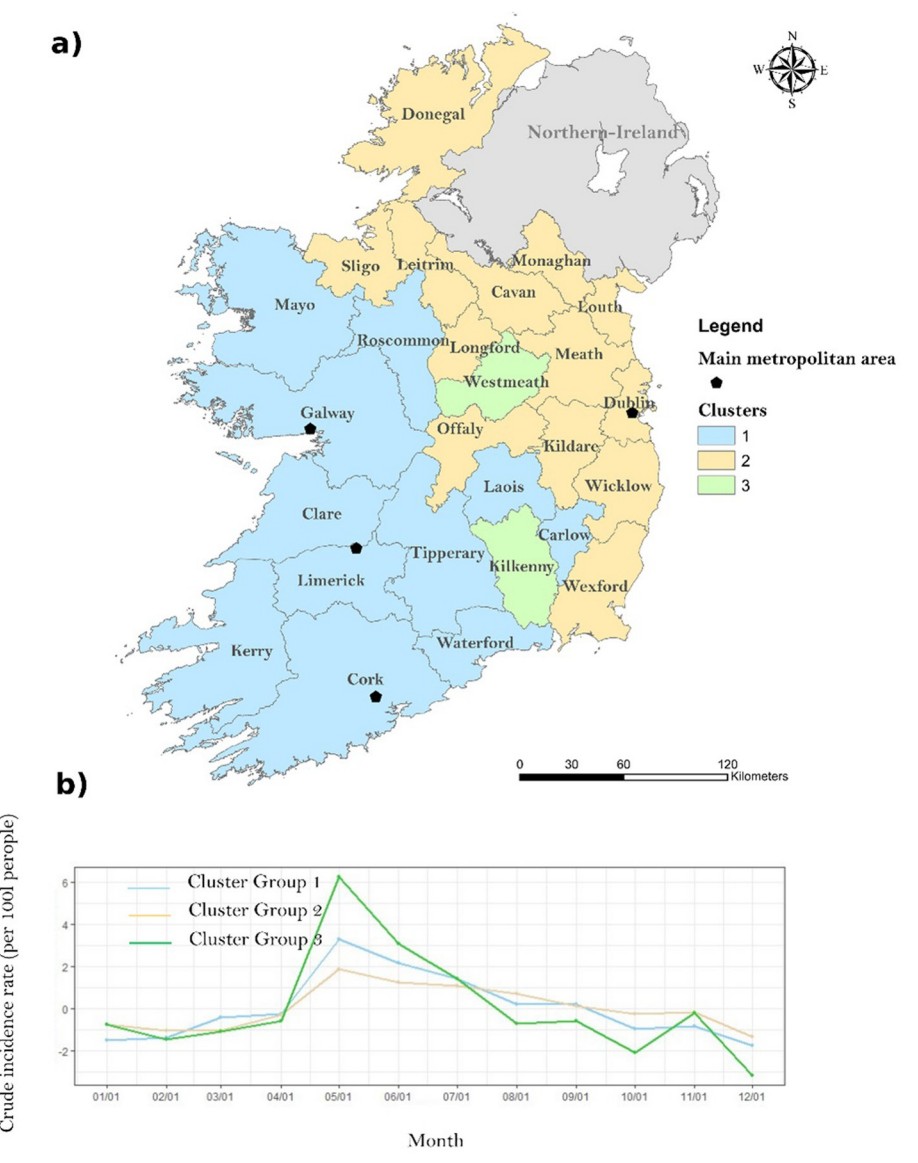

**Fig 6.** (a) Seasonal clustering of campylobacteriosis by county, 2011–2018 –(b) Seasonal variation of campylobacteriosis incidence rate (per 100k people) for each clusters.

were located around the urban centres of Dublin, Galway, Cork and Limerick, along with counties Kilkenny, Westmeath, Tipperary, Wicklow, Waterford and Wexford Conversely, cold spots of infection were mainly concentrated in rural areas and particularly on the northern part of the country, adjacent to the Northern Irish border (i.e., counties Donegal, Roscommon, Meath, Leitrim, Louth and Monaghan), and on the West and South-West of the island (i.e., counties Mayo, Sligo and Kerry). The highest proportion of SAs categorised as cold-spots were reported in counties Louth (85%. of total SAs), Meath (74.7%) and Longford (72.8%).

**Table 3. Seasonal clusters characteristics (case number, gender, age, settlement type).**

| Clusters | 1 | 2 | 3 | Total |
|---|---|---|---|---|
| **General characteristics (N = 20,355)** | | | | |
| Total cases (% of cases) | 8,964 (44%) | 10,264 (50.4%) | 1,127 (5.6%) | 20,355 |
| Mean yearly CIR | 68.53 | 46.56 | 78.06 | 64.38 |
| Mean monthly CIR | 4.92 | 4.09 | 6.24 | 5.08 |
| Total Population (2016 Census–In Million) | 1.635 (35.8%) | 2.756 (60.3%) | 0.180 (3.9%) | 4.571 |
| Number of counties | 10 | 14 | 2 | 26 |
| **Gender (N = 20,355)** | | | | |
| M (% of cases) | 5,006 (55.8%) | 5,508 (53.7%) | 607 (53.9%) | 11,121 (54.6%) |
| F (% of cases) | 3,958 (44.2%) | 4,756 (46.3%) | 520 (46.1%) | 9,234 (45.4%) |
| **Age of cases (N = 20,355)** | | | | |
| Mean age (years) | 30.8 | 34.7 | 30.4 | 31.96 |
| $\leq 5$ y. | 2,407 (26.8%) | 1,910 (18.6%) | 310 (27.4%) | 4,627 (22.7%) |
| $> 5$ & $< 65$ y. | 5,240 (58.4%) | 6,777 (66.1%) | 655 (58%) | 12,672 (62.3%) |
| $\geq 65$ y. | 1,320 (14.6%) | 1,571 (15.3%) | 165 (14.6%) | 3,056 (15%) |
| **Settlement type (N = 14,338)** | | | | |
| Rural (% of cases) | 1,545 (17.2%) | 1,003 (9.8%) | 191 (16.9%) | 2,739 (13.5%) |
| Commuter (% of cases) | 1,201 (13.4%) | 644 (6.3%) | 210 (18.6%) | 2,055 (10.1%) |
| Urban (% of cases) | 3,370 (37.6%) | 5,856 (57.1%) | 318 (28.2%) | 9,544 (46.9%) |
| No geographical information | 2,848 (31.8%) | 2761 (26.9%) | 408 (36.2%) | 6,017 (26.9%) |

Urban areas were positively associated with the presence of an infection spatial hotspot (aOR: 3.34, 95% CI: 3.09–3.61), while negative associations were found with rural (aOR: 0.25, CI 95% 0.22–0.27) and commuter areas (aOR: 0.60, CI 95% 0.6–0.74) (Table 4).

## 3.5 Space-time cluster frequency mapping

The developed cluster frequency map for campylobacteriosis across the ROI is shown by Fig 8 while Table 5 presents the SA-level cluster frequency distribution. Results indicate that 76.5% of total SAs reported at least one space-time cluster from 2011 to 2018, with an overall mean of 1.5 cluster frequency per SA. A maximum of 6 (i.e., meaning a space-time cluster was reported within this area 6 years from 2011 to 2018) was obtained for a single Small Area, located in county Tipperary. Identified spatio-temporal hotspots of infection characterised by (high cluster frequencies) (i.e., > national mean, $\geq 2$) were primarily situated along a South-West to South-East axis from County Cork to County Kilkenny along with counties Dublin and Wicklow. Secondary hotspots of infection with areas having a cluster frequency equal to 1 (i.e., infection hotspot present during 1 year), were reported within the periphery of the aforementioned primary hotspot areas, in addition to County Westmeath (midlands) and in counties Cavan, Sligo and Leitrim (north). Space-time clusters were typically identified from May to July with a maximum occurrence during May (n = 15). No significant space-time clusters were identified from November to February.

Rural SAs were less likely to exhibit the presence of a space-time cluster frequency (aOR: 0.22, CI 95% 0.20–0.23) (Table 6). Likewise, commuter areas were 0.82 times less likely to report a space-time cluster. Conversely, urban areas were positively associated with the presence of cluster frequency $\geq 1$ (aOR: 3.96. CI 95% 3.71–4.23) and $\geq 2$ (aOR: 2.74, CI 95% 2.57–2.93).

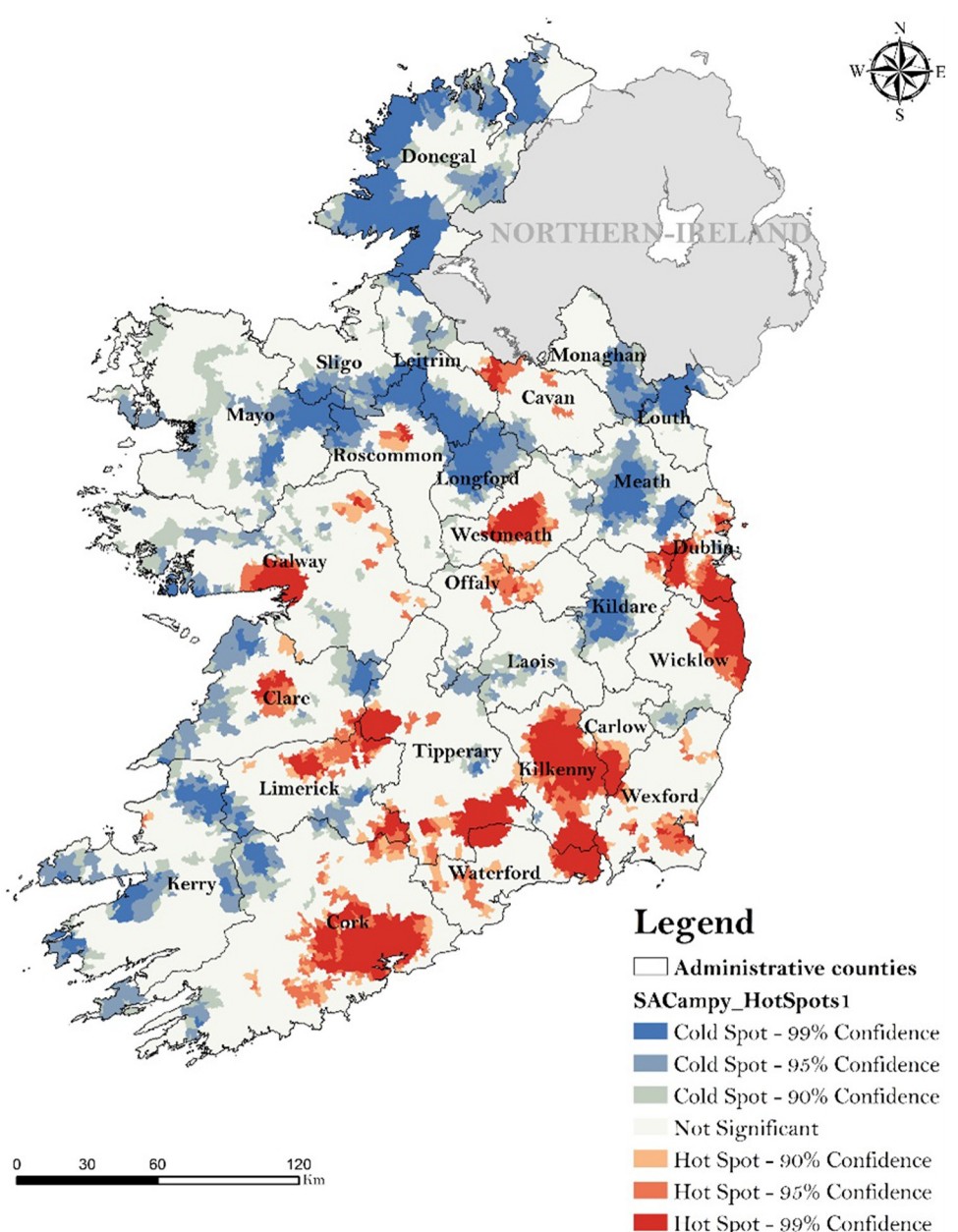

**Fig 7. Results from the spatial autocorrelation (GetIs OrdGi) of campylobacteriosis in the Republic of Ireland (2011–2018)–geo-linked dataset (N = 14,338).**

## 4. Discussion

The current study sought to assess the spatio-temporal epidemiology of campylobacteriosis in the Republic of Ireland from 2011 to 2018 (n = 20,391) using a series of novel geospatial analysis techniques such as seasonal clustering and space-time cluster frequency mapping.

Data distribution revealed a higher incidence of campylobacteriosis among children aged under 5 years old, representing 22.7% of total sporadic cases from 2011 to 2018 (Table 1). Study results mirror previous findings from multiple scientific studies highlighting the highest incidence of campylobacteriosis among children [21–23]. The high incidence of

**Table 4. Results from fisher exact tests and presence of hot/cold spots by main settlement areas–NS: Non significant p-value.**

|  | SA | CI 5% | aOR | CI 95% | P-Val |
|---|---|---|---|---|---|
| **Rural** | | | | | |
| Hotspot | 522 | 0.22 | 0.25 | 0.27 | <0.001 |
| Cold spot | 1,085 | 1.80 | 1.96 | 2.13 | <0.001 |
| **Commuter** | | | | | |
| Hotspot | 504 | 0.60 | 0.67 | 0.74 | <0.001 |
| Cold spot | 514 | 1.42 | 1.58 | 1.76 | <0.001 |
| **Urban** | | | | | |
| Hotspot | 4,034 | 3.09 | 3.34 | 3.61 | <0.001 |
| Cold spot | 1,264 | 0.41 | 0.44 | 0.48 | <0.001 |

campylobacteriosis among children under 5 years old is likely explained by a lower immunological profile within this age group, with the infection tending to become increasingly asymptomatic among higher age groups [23]. A study by Havelaar et al. [24] suggests that children and individuals having previously experienced a primary exposure to the pathogen are more likely to develop antibodies and therefore less likely to develop symptoms during a secondary exposure. A relatively high incidence was observed amongst adults aged 20 to 35 years old (n = 4,175, 20.5% of total cases, Table 1). Similar observations have been reported by Kuhn et al. [25] in Denmark, Louis et al. [21] in England and Wales, and Schielke et al. in Germany [4]. Increased levels of infection among young adults (20–35) has been attributed to risky cooking and eating behaviours upon moving out of the family home [26], in concurrence with a potentially affected immunological status due to this change of environment [25]. Findings revealed that paediatric infections were more likely to occur in commuter and rural areas (S1 and S2 Tables), mirroring previous findings from Nichols et al. [27]. The relative prevalence of paediatric infection in rural areas may be explained by higher exposures to environmental risk factors such as direct livestock exposure or contaminated water sources [24–26]. Likewise, the prevalence of cases among aging populations (>65 years) was positively associated with rural living (aOR: 1.25, CI 95% 1.11–1.4, Table 2), potentially indicative of higher environmental exposures, lower access to healthcare facilities or unique dietary patterns. For example, a recent study from New-Zealand identified unpasteurized milk consumption, more common in rural areas due to the farm proximity, as a potential driver of campylobacteriosis [28].

The higher overall incidence of male cases was particularly pronounced among the younger age groups (<5years), with male cases statistically more likely within the paediatric sub-population (aOR: 1.27, CI 95% 1.18–1.35, Table 2). Higher incidence rates of campylobacteriosis were consistently reported among males, and particularly among male children under 4 years old in Australia from 1998 to 2013 [29]. Similar observations were made by Schielke et al. in Germany [4] and Kuhn et al. [25] in Denmark. Considering that gender-specific behaviours are typically not involved among younger age groups, the higher incidence associated with male patients may be related to physiological differences resulting in a higher susceptibility in terms of transmission/contraction and severity [22, 30]. Similarly, as shown (Table 2), higher likelihoods of contracting campylobacteriosis were noted amongst males residing in rural (aOR: 1.005, CI 95% 0.92–1.09) and commuter (aOR: 1.173, CI 95% 1.07–1.29) areas. This study finding may reflect a higher risk of transmission via direct animal contact among the rural and male sub-population. While occupational status was not highlighted as a risk factor for campylobacteriosis in Ireland by Danis et al. [8], Green et al. found that the male population under 19 years of age in rural areas of Manitoba (Canada) and engaging in agricultural

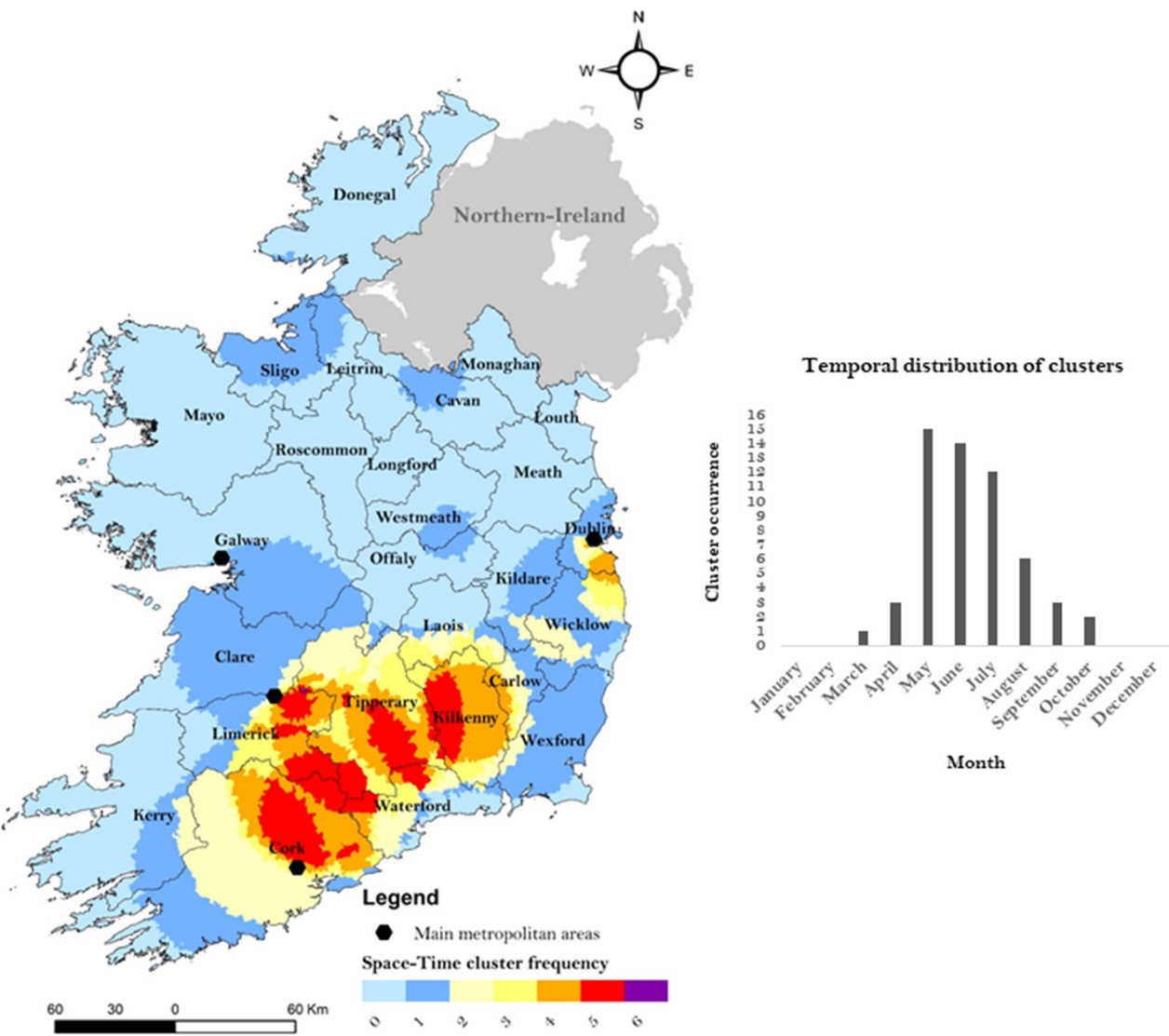

**Fig 8. Space-time cluster frequency mapping of campylobacteriosis (2011–2018) in the Republic of Ireland.**

practices, represented the highest risk group for campylobacteriosis [10]. Likewise, over 73% of the current agricultural labour force in the ROI (n = 278,600) identify as male [31].

Seasonal decompositions revealed an infection peak from April to August, confirming the higher incidence of infection during summer months reported across the majority of Europe [32]. The peak of infection was reported during May (2.54 cases/100,000), and therefore slightly earlier than in the United-Kingdom where the infection typically peaks in June [33, 34]. The summer peak of infection is traditionally associated with higher levels of outdoor

**Table 5. Space-time cluster frequency distribution across small areas.**

| Settlement types | Cluster frequency | | | | | | | |
|---|---|---|---|---|---|---|---|---|
| | 0 | 1 | 2 | 3 | 4 | 5 | 6 | Total |
| Total (% Total SAs) | 6,252 (33.5) | 5,320 (28.5) | 2554 (13.7) | 1,988 (10.7) | 1,846 (9.9) | 680 (3.6) | 1 (-) | 18641 (100) |

**Table 6. Fisher Tests results between presence/absence of a space-time cluster $\geq 1$ and cluster frequency $\geq 2$ between main settlement types -[NS]: Non significant p-value.**

| Settlement types | CI 5% | aOR | CI 95% | P-Val |
|---|---|---|---|---|
| **Space-time cluster $\geq 1$** | | | | |
| Rural | 0.20 | 0.22 | 0.23 | $\leq 0.001$ |
| Commuter | 0.75 | 0.82 | 0.9 | $\leq 0.001$ |
| Urban | 3.71 | 3.96 | 4.23 | $\leq 0.001$ |
| **Space-time cluster frequency $\geq 2$** | | | | |
| Rural | 0.25 | 0.27 | 0.3 | $\leq 0.001$ |
| Commuter | 0.84 | 0.92 | 1.01 | 0.069[NS] |
| Urban | 2.57 | 2.74 | 2.93 | $\leq 0.001$ |

activity/recreation, exposure (contact) to farm animals and wildlife, higher risk of consuming undercooked meats due to seasonal cooking habits (i.e., barbecues), and consumption of untreated water [32, 35]. An overarching increase was observed in the decomposed long-term trend of infection over the 8-year study period (Fig 5C). A study by O'Connor et al. [9]. has previously reported a similar increase in campylobacteriosis notifications from 2011 to 2016 across Ireland, while the ECDC highlight a subsequent decrease across Europe including the ROI from 2017 to 2020 (with an abnormal low level of cases during 2020 likely associated with the COVID-19 pandemic), followed by a slight increase again in 2021 [2]. Interestingly, the year 2018 reported the highest number of cases during the study period (61.96 cases/100,000) with an abnormal excess (residual) of cases identified between March and July 2018 (Fig 5D). This notably elevated level of incidence was likely driven by the 2018 summer drought event (particularly during May/June), which was the most severe hydrological drought experienced in the country since 1981 [36]. Similarly, high residuals were identified during January 2016 (Fig 5D), which synchronises with one of the most severe and nationally extensive flood events reported within the ROI [37]. These findings point to the existence of potentially strong relationships between extreme weather events and campylobacteriosis in the ROI. While the occurrence of heatwaves has been linked with a decrease in campylobacteriosis incidence due to a shorter pathogen life expectancy [7], an increased incidence has previously been associated with drier, sunnier conditions due to higher rates of food contamination (i.e., barbecues), in concurrence with potentially higher risks of groundwater contamination [27, 38]. Similarly, heavy rainfall events have been reported to trigger an increase of sporadic (i.e., primary cases) campylobacteriosis and potential outbreaks due to contamination of public water systems and a longer survival capacity of the bacteria in wet conditions [38, 39] Spatial clustering identified three temporal clusters of counties characterised by similar seasonal signals (Fig 6), with the primary geographic delineations being between counties located on the East and North (Cluster 2) and counties located in the West (Cluster 1). Counties from Cluster 2 (West) were characterised by a higher rate of infection (Table 3) and a slightly earlier case peak (February/March, Fig 6B). Cluster 1 (East) was characteristically urban/commuter, as influenced by the presence of the Greater Dublin Area (accounting for 61.4% of total urban SAs), with cases notified within this Cluster approximately 3 times more likely to be associated with an urban SA (S2 Table). A recent study by Djennad et al. [14] observed a similar campylobacteriosis lag within the metropolitan areas of Birmingham and London in the UK when compared to Wales and the North/East of England. The authors hypothesised that this may be linked with spatially specific socio-demographic profiles, risk exposures and weather patterns [14]. A significant higher mean case age was reported within Cluster 2 (S1 Table), with a markedly lower proportion of paediatric infections (Table 3). Strachan et al. [40] previously observed an earlier

infection peak amongst children in rural areas of Scotland, pointing to distinct age-specific transmission pathways with a higher risk of environmental contamination (i.e., direct animal contact, ruminant or wild bird, consumption of contaminated water) rather than foodborne transmission among the youngest sub-populations. Similarly, the disparity between Cluster 2 and 3 may be reflective of fluctuating weather patterns across the ROI, in addition to potentially differing dietary patterns vis. settlement type. In addition to the East/West geographical pattern (Clusters 1 and 2), seasonal clustering also identified counties Kilkenny and Westmeath as a specific Cluster (Cluster 3, Fig 6A). This Cluster was characterised by a significantly higher incidence rate (78 cases/100,000 per annum), a particularly marked infection peak in May and secondary peaks of infection in November and January (Fig 6B).

Spatial autocorrelation (Fig 7) revealed the presence of hotspots on a South-East axis (from Cork and Limerick counties to Dublin County), with this region (counties Cork, Tipperary, Kilkenny and Limerick) also clearly highlighted by space-time cluster frequency mapping (Fig 8). The presence of infection hotspots within this geographic area might be linked with specific agricultural practices; dairy farming has previously been highlighted as a risk factor of human campylobacteriosis due to higher bacterial exposure [41]. Accordingly, the southern part of the island is the primary location of intensive dairy farming in the ROI [42], while the Northern part of the country is characterised by mixed agricultural activities such as small dairy farming, arable farming, or cattle and sheep grazing [43] Static clustering also revealed the presence of hotspots around the main urban centres, namely Dublin, Cork, Limerick and Galway cities (Fig 8). As such, urban areas were found to be 6.5 times more likely to be situated within an infection hotspot (Table 4). Similarly, a positive association was found between urban living and the presence of a space-time cluster (aORs: 3.96, Table 6). These results highlight the concentration of infection hotspots within densely populated areas. Conversely, numerous studies have reported a more significant association between rural areas including Levesque et al. [44] in Canada, Miller et al. [45] in Scotland and Fitzenberg et al. [46] in Germany, albeit a positive association between campylobacteriosis and urban living was identified by Van Hees et al. [47] in Netherlands. The concentration of infection hotspots in urban areas of Ireland might be linked with higher rates of foodborne contamination linked with consumption of ready-to-eat foods and restaurants [47] and particularly with respect to cases within the intermediate age groups (Table 2). This exposure route has previously been identified as a primary risk factor for campylobacteriosis in Ireland [8]. The authors notably revealed that food consumption from take-away restaurants was one of the key driver for campylobacteriosis in Ireland, likely associated with cross-contamination due to poor hygiene practice [8]. In contrast, infection cold spots (Fig 7) and low cluster frequencies (Fig 8) were localised in Western (i.e., Kerry, Mayo) and Northern areas (i.e., Donegal, Sligo, Leitrim, Louth) potentially reflecting local agricultural practices with a lower concentration in livestock and dairy farming industry [43] and local geography, characterised by a general higher annual precipitations [48]. While heavy rainfall events have been associated with increased incidence rate of campylobacteriosis [7, 49], significantly wetter conditions may indicate lower pathogen exposure, due to change of cooking habits (i.e., less barbeques) and lower rate of outdoor activities. Overall, space-time scanning identified 23 annual space-time clusters of sporadic infection from 2011 to 2018, raising significant concerns for the public health and food safety authorities of Ireland. This, in concurrence with the high incidence of campylobacteriosis notified during the study period (55 cases/100,000), leaves little doubt that the ROI represents a relatively high-risk region for contracting campylobacteriosis. This has also been associated with traditional eating behaviours such as high chicken consumption, reported to be part of the diet of approximately 70% of the Irish population [8]. A study from Madden et al. [50] that *campylobacter spp.* was prevalent in 84.3% of raw chicken samples in retail sales in Ireland. Similarly,

Consumption of undercooked chicken and contaminated ready-to-eat (RTE) foods during the summer months have been identified as important risk factors for sporadic Campylobacter spp. infection in both the Republic of Ireland and Northern Ireland [8].

## 5. Conclusion

The current study explored the spatio-temporal characteristics of campylobacteriosis cases in Ireland from 2011 to 2018. Research findings revealed the existence of distinct infection pathways across age groups, with younger and older groups associated with rural areas while the intermediate age group was linked with urban living. This result suggest the need of developing geographically age-specific surveillance strategies. Seasonal decomposition and seasonal clustering pointed out the existence of spatially specific temporal patterns of infection in Ireland, with a higher and stronger seasonal signal reported in the Western part of the country. Similarly, spatial autocorrelation and space-time scanning techniques pointed out clear evidence of campylobacteriosis infection hotspots across the country, mostly located around the main urban areas of the country (i.e., Dublin, Cork, Limerick) and in the Southern part of the island. While additional analyses are required to point out the exact influence of local risk factors, these results point out spatio-temporally specific socio-demographic (e.g., dietary types, cooking habits) and environmental profiles (e.g., local weather, agricultural practices), therefore highlighting the need for initiating spatio-temporally targeted health management strategies.

## Supporting information

**S1 Table. Tukey pairwise comparison test results for clusters and age.**
(DOCX)

**S2 Table. Fisher tests results for clusters and main settlement types (rural, urban, commuter areas).**
(DOCX)

**S3 Table. Percentage of total SAs identified as hot/cold spots and statistically unsignificant for each county.**
(DOCX)

## Author Contributions

**Data curation:** Coilín ÓhAiseadha, Patricia Garvey.

**Formal analysis:** Martin Boudou.

**Investigation:** Martin Boudou.

**Methodology:** Martin Boudou, Paul Hynds.

**Project administration:** Paul Hynds.

**Supervision:** Patricia Garvey, Jean O'Dwyer, Paul Hynds.

**Validation:** Coilín ÓhAiseadha, Jean O'Dwyer.

**Visualization:** Martin Boudou.

**Writing – original draft:** Martin Boudou.

**Writing – review & editing:** Jean O'Dwyer, Paul Hynds.

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
