## [Decision Letter · Decision Letter 0]

18 Jul 2023

PONE-D-23-16493An ecological study of spatiotemporal Dynamics and Drivers of Domestically Acquired Campylobacteriosis in Ireland, 2011–2018PLOS ONE

  Dear Dr. Boudou,

Thank you for submitting your manuscript to PLOS ONE. After careful consideration, we feel that it has merit but does not fully meet PLOS ONE’s publication criteria as it currently stands. Therefore, we invite you to submit a revised version of the manuscript that addresses the points raised during the review process.

ACADEMIC EDITOR: Please address all of the reviewers' comments. 

We look forward to receiving your revised manuscript.

Kind regards,

Csaba Varga, DVM MSc PhD

Academic Editor

PLOS ONE

“-Initial of the funded author: PH

-Grant number: 2019-CCRP-MS.62

-Funder name: Environmental Protection Agence Ireland

- https://www.epa.ie/

- No”

4. We note that Figures 1, 3, 6, 7 and 8 in your submission contain [map/satellite] images which may be copyrighted. All PLOS content is published under the Creative Commons Attribution License (CC BY 4.0), which means that the manuscript, images, and Supporting Information files will be freely available online, and any third party is permitted to access, download, copy, distribute, and use these materials in any way, even commercially, with proper attribution. For these reasons, we cannot publish previously copyrighted maps or satellite images created using proprietary data, such as Google software (Google Maps, Street View, and Earth). For more information, see our copyright guidelines: http://journals.plos.org/plosone/s/licenses-and-copyright.

1. You may seek permission from the original copyright holder of Figures 1, 3, 6, 7 and 8 to publish the content specifically under the CC BY 4.0 license. 

Reviewers' comments:

Reviewer's Responses to Questions

**Comments to the Author**

1. Is the manuscript technically sound, and do the data support the conclusions?

Reviewer #1: Partly

Reviewer #2: Partly

2. Has the statistical analysis been performed appropriately and rigorously? 

Reviewer #1: No

Reviewer #2: Yes

3. Have the authors made all data underlying the findings in their manuscript fully available?

Reviewer #1: No

Reviewer #2: No

4. Is the manuscript presented in an intelligible fashion and written in standard English?

Reviewer #1: Yes

Reviewer #2: Yes

5. Review Comments to the Author

Reviewer #1: This study aims to examine Campylobacteriosis in Ireland from 2011-2018 using a variety of spatial science techniques. It is generally very well written but suffers from mainly descriptive analyses. I have provided suggestions for improvement below that will make this a stronger study with the available data, using more rigorous statistical techniques.

Main concerns:

1. Geostatistics are a subfield of spatial statistics that explicitly deal with predicting unknown values. No geostatistical methods are employed in this paper. Please replace all instances of "geostatistics" with "spatial statistics".

2. The introduction should discuss more literature outside of Ireland and the UK. The use of spatial statistics to study the bacteria in the literature should also be mentioned. I also suggest explicitly stating the gaps that your study is filling.

3. Was the hotspot analysis computed with cases or rates? I suggest rates since cases can be a function of population. Good to show hotspots of rates rather than cases in epidemiological settings.

4. I think you missed an opportunity to provide stronger results by adjusting for key covariates in the SaTScan analysis and conducting formal regression modeling rather than relying on correlation tests. Since you have a nice set of covariates, it would be much stronger to detect space-time clusters of disease after adjusting for your potential covariates, which can be adjusted for directly in the model, or even better - using regression fitted values from a Poisson GLM. I think this will make the paper much stronger and get away from a mostly descriptive analysis.

Minor comments:

-Please report confidence intervals next to the aORs in the abstract.

-Please add CIs in text when you mention an aOR.

Reviewer #2: Thank you for the opportunity to review this manuscript. The presented findings are interesting and would likely be of interest to local public health agencies. The manuscript could be further strengthened through the provision of some additional background information and discussion, and minor clarification of methods/results, as outlined below for consideration:

Introduction

The 2009 study by Danis et al. also noted that contact with small ruminants (sheep) increased the odds of subsequent campylobacteriosis. Suggest including some information in the discussion regarding previously identified risk factors and the usual reservoir for the bacterium for the reader who may be unfamiliar with this pathogen. It would be helpful to contextualize the findings by providing some information on the prevalence and geographic distribution of animal reservoirs (cattle/small ruminants etc.) in the ROI for the reader who is unfamiliar with the ROI. Additionally, if individuals in rural areas are more likely than their urban counterparts to drink raw (unpasteurized) milk, this could provide an additional means of animal-related foodborne exposure in rural areas, as has been found to be the case in other developed countries.

Methods

Clarify if all cases included in analyses were those considered to be sporadic (i.e., were those linked to known outbreaks excluded? If so how many cases/outbreaks were there in the dataset?). There were several known outbreaks of campylobacteriosis in the ROI during the study period (per https://www.frontiersin.org/articles/10.3389/fcimb.2022.979055/full)

Clarify how many cases were lost during data cleaning and why (n=6,053? it’s not clear if cases were lost at the SA linkage step and why – i.e., lack of address data? or if data were excluded for other reasons, e.g. lack of individual data for other variables examined)

Results

The authors note that per the data in Table 1, the intermediate (5-65 year) and older age groups were more likely to be female, however, Figure 2 appears to show that for all age groups other than those aged 24-30, 30-35 and 85-90, cases were more likely to be male. Suggest clarifying why age was collapsed into such broad groupings (particularly the intermediate category) as potentially significant differences between age groups (as shown in Figure 1) seem to have been lost through aggregation of cases in the intermediate age group, potentially changing the direction of association. From the data presented in Table 2, it appears that the data for the intermediate age group could have been separated into three or four different categories/groupings to create similar sized age groupings for analyses. Assessment of age via smaller groupings may have produced different findings, more similar to Figure 2.

For those cases that were not able to be geolinked, were these randomly distributed or did this result in the complete loss of certain areas for spatial analyses?

It would be helpful for the international reader if the authors could clarify what months Irish summer occurs in. Winter is noted as being Dec-Feb, however the authors refer to May as being during summer – in other countries in the northern hemisphere May is considered to be in spring (typically March-May). Similarly, Figure 4 appears to show that May is a summer month and that February occurs during Spring (not winter as stated in the results section). Suggest using consistent categorizations for season throughout the manuscript.

In Figure 4, suggest clarifying the case numbers in the legend as (n=3,015) etc. instead of (3,015n) so this is clearer for the reader.

Suggest clarifying in the results section which method produced each set of results. i.e., were the results for seasonal clustering in section 3.3. obtained through space-time scans? Were the results of each methodology similar where applicable or did any methods produce conflicting findings?

Discussion:

In the results section the authors note that a positive association was found between female cases and urban settlements, and male cases were more likely to occur in commuter regions and rural areas. Suggest providing additional discussion re: why this might be. I.e. are female cases potentially more likely to be food-related and male cases more likely to be animal related if males are more likely to work in agricultural occupations? Similarly, food consumption patterns among commuters may differ from non-commuters – e.g., increased tendency to purchase fast food. As the authors noted, FSA Ireland and other studies have previously found consumption of RTE food to be a risk factor for campylobacteriosis.

As above, it would be helpful to provide some additional discussion re: potential agricultural explanations for identified hot and cold spots. For the reader who is unfamiliar with the ROI, it would also be interesting to provide some discussion re: how the location of hot and cold spots relates to the distribution of reservoir agricultural species (e.g., cattle, small ruminants), contact with which is a known risk factor for infection. Previous studies have found that prolonged ongoing exposure can induce immunity to infection, which could be a potential explanation for the

identification of cold spots in rural/agricultural areas (if applicable), particularly in older age groups.

The authors note that high rates in children may be due to increased susceptibility, while older age groups may be more likely to be symptomatic – this appears to indicate that observed incidence in the <5 age group may be an underestimate if children are more likely than older age groups to be asymptomatic or to only have mild illness (and less likely to be tested).

As a note: suggest defining ROI at first use and using this consistently throughout the manuscript

6. PLOS authors have the option to publish the peer review history of their article (what does this mean?). If published, this will include your full peer review and any attached files.

Reviewer #1: No

Reviewer #2: No

---

## [Author Response · Author response to Decision Letter 0]

11 Aug 2023

An Ecological Study of the Spatiotemporal Dynamics and Drivers of Domestically Acquired Campylobacteriosis in Ireland, 2011–2018

Reviewer Comments and Author Responses

Reviewer #1: This study aims to examine Campylobacteriosis in Ireland from 2011-2018 using a variety of spatial science techniques. It is generally very well written but suffers from mainly descriptive analyses. I have provided suggestions for improvement below that will make this a stronger study with the available data, using more rigorous statistical techniques.

The authors would like to thanks reviewer #1 for his remarks. We believe we have address all his comments, contributing to improve the quality of the manuscript.

Reviewer #1 – 1°. Geostatistics are a subfield of spatial statistics that explicitly deal with predicting unknown values. No geostatistical methods are employed in this paper. Please replace all instances of "geostatistics" with "spatial statistics".

The authors agree that the term “geostatistics” was not applied correctly, and accordingly, the term “geostatistics” has been replaced by “spatial statistics” throughout the manuscript file. 

Reviewer #1 – 2°. The introduction should discuss more literature outside of Ireland and the UK. The use of spatial statistics to study the bacteria in the literature should also be mentioned. I also suggest explicitly stating the gaps that your study is filling.

The authors agree that audience understanding may be improved via inclusion of more detail around the use of spatial statistics to study bacterial infections. Accordingly, additional information relating to the use of spatial statistics in epidemiological studies of campylobacteriosis have been added to the original introduction (See below). With regards to the gaps the current study is filling, to date, no national scale study of geo-referenced cases of campylobacteriosis or the spatio-temporal characteristics of infection in the Republic of Ireland have been undertaken and/or published. The text provided below has been added to make this clear for the reader: 

“While both studies present findings pertaining to the likely risk factors for contracting campylobacteriosis in the ROI, to date, no comprehensive study has been undertaken to examine the spatio-temporal characteristics of infection at a national scale, thus effectively minimising any potential spatial or temporal biases. Spatial statistics and time-series analysis have previously been employed to identify the primary spatio-temporal drivers of campylobacteriosis. For example, Green et al. (2020) employed spatial scan statistics to identify high and low rate clusters of campylobacteriosis from 1996 to 2004 in Manitoba, Canada. Similarly, Marek et al. (2015), used spatio-temporal Kriging spatial interpolation and spatial scan statistics to describe the spatio-temporal patterns of campylobacteriosis in Czech Republic (2008-2012).”

Reviewer #1 – 3°. Was the hotspot analysis computed with cases or rates? I suggest rates since cases can be a function of population. Good to show hotspots of rates rather than cases in epidemiological settings. 

The authors would like to thank the reviewer for what is an important question/consideration, and we wholly agree that hotspots should be based on CIRs as opposed to case number to avoid spatial biases and/or ecological fallacy due to background human population density. Hotspot analyses were performed using crude incidence rates per 100k people throughout this work, and this has now been made clear (See below): 

“In the current study, crude incidence rate per 100k people for each SA was used as the primary input for identification of hot/cold spots of laboratory-confirmed campylobacteriosis across Ireland”

Notwithstanding, based on the reviewers comment, the authors undertook a comprehensive “double check” of all presented analyses and findings, and realised some mistakes appeared in Tables 4 & S3, relating to the associations between spatial autocorrelation with settlement types and the percentages of hot/cold-spots by counties. These tables have been double-checked and amended within the results and discussion sections.

Reviewer #1 – 4°. I think you missed an opportunity to provide stronger results by adjusting for key covariates in the SaTScan analysis and conducting formal regression modelling rather than relying on correlation tests. Since you have a nice set of covariates, it would be much stronger to detect space-time clusters of disease after adjusting for your potential covariates, which can be adjusted for directly in the model, or even better - using regression fitted values from a Poisson GLM. I think this will make the paper much stronger and get away from a mostly descriptive analysis.

Thank you for another insightful question, and one we would have asked ourselves. Generalised Linear Modelling using a Poisson link function was initially employed by the authors in an attempt to identify the main spatial risk factors of campylobacteriosis (In fact, this was the first statistical approach used). However, due to the “thin” spatial resolution of the CSO Small Area (N = 18,641), it was not possible to achieve good model accuracy (primarily due to a heavily zero- and one-inflated dataset i.e., very very few SAs associated with >1 case). Similarly, a “restricted modelling” approach with SatScan (i.e., age groups delineation) was tested. This approach encountered the same issues as the aforementioned Poisson GLMs, via very significantly decreasing the number of identified space-time clusters with RR>1, required to understand potential variations of campylobacteriosis dynamics between and within specific age groups. The authors therefore elected not exclude these approaches from the current manuscript. With regard to the “descriptive” nature of the study, the authors understand and respect the Reviewers perspective/opinion. However, we consider this article to be a “foundation” paper related to campylobacteriosis in the Republic of Ireland i.e., future articles will build on findings presented here, with the cluster recurrence index employed as a fundamental dependant variable for ensuing modelling. For example, further analyses are currently underway to assess spatio-temporal variations of disease incidence and the cluster recurrence index linked with weather and seasonality in Ireland. 

Minor comments:

Reviewer #1 – 5°. Please report confidence intervals next to the aORs in the abstract. 

Amended as requested throughout the manuscript.

Reviewer #1 – 6°. Please add CIs in text when you mention an aOR. 

Amended as requested throughout the manuscript.

Reviewer #2: Thank you for the opportunity to review this manuscript. The presented findings are interesting and would likely be of interest to local public health agencies. The manuscript could be further strengthened through the provision of some additional background information and discussion, and minor clarification of methods/results, as outlined below for consideration:

The authors would like to thanks reviewer #2 for his time and expertise on our work. We have addressed each comment individually. 

Reviewer #2 – 1°. The 2009 study by Danis et al. also noted that contact with small ruminants (sheep) increased the odds of subsequent campylobacteriosis. Suggest including some information in the discussion regarding previously identified risk factors and the usual reservoir for the bacterium for the reader who may be unfamiliar with this pathogen. It would be helpful to contextualize the findings by providing some information on the prevalence and geographic distribution of animal reservoirs (cattle/small ruminants etc.) in the ROI for the reader who is unfamiliar with the ROI. 

Dear reviewer, unfortunately, the study from Danis et al. (2009) does not integrate any spatial information, making it difficult to relate study findings to the current analysis. Additional information related to geographic distribution of animal reservoirs in the ROI was however added within the discussion. See response Reviewer #2 – 12°.

Reviewer #2 – 2°. Additionally, if individuals in rural areas are more likely than their urban counterparts to drink raw (unpasteurized) milk, this could provide an additional means of animal-related foodborne exposure in rural areas, as has been found to be the case in other developed countries. 

Response Reviewer #2 – 27°. Dear reviewer, thank you for providing us with this comment/observation. Dairy farming was indeed mentioned as one likely driver for the spatial distribution of campylobacteriosis hotspots in Ireland within the manuscript (See below):

“The presence of infection hotspots within this geographic area might be linked with specific agricultural practices; dairy farming has previously been highlighted as a risk factor of human campylobacteriosis due to higher bacterial exposure [37], with the dairy industry in the ROI primarily concentrated in the southern region of the island [38].”

Additional information related to rural areas were added to the discussion:

“The relative prevalence of paediatric infection in rural areas may be explained by higher exposures to environmental risk factors such as direct livestock exposure or contaminated water sources [22, 23, 25]. Likewise, the prevalence of cases among aging populations (>65 years) was positively associated with rural living (aOR: 1.25, CI 95% 1.11 – 1.4, Table 2), potentially indicative of higher environmental exposures, lower access to healthcare facilities or unique dietary patterns. For example, a recent study from New-Zealand identified unpasteurized milk consumption, more common in rural areas due to the farm proximity, as a potential driver of campylobacteriosis (Davys et al., 2020).”

Reviewer #2 – 3°. Clarify if all cases included in analyses were those considered to be sporadic (i.e., were those linked to known outbreaks excluded? If so how many cases/outbreaks were there in the dataset?). There were several known outbreaks of campylobacteriosis in the ROI during the study period (per https://www.frontiersin.org/articles/10.3389/fcimb.2022.979055/full)

Response Reviewer #2 – 3°. Dear reviewer, yes, we can confirm that only sporadic and primary cases of infection where employed for analyses to avoid detection of these outbreaks by SaTScan (which would significantly bias any analyses of disease dynamics). Further explanation have added in the method section. 

“Cases associated with recent international travel (N = 81) and secondary cases associated with outbreaks (N = 159) were excluded from the current study”

Reviewer #2 – 4°. Clarify how many cases were lost during data cleaning and why (n=6,053? it’s not clear if cases were lost at the SA linkage step and why – i.e., lack of address data? or if data were excluded for other reasons, e.g. lack of individual data for other variables examined) 

Response Reviewer #2 – 4°. The exact number of non-geographically linked cases have been added to the methods section, along with further explanation. This issue, frequently encountered and described within the field of spatio-temporal epidemiology and ecological studies, arises from an absence of complete and/or legible address details during the surveillance data collection. The authors, not being involved within that process for data sensitivity reasons were therefore unable to address it. Proportion of missing cases at the SA level however exhibited a non-systematic (random) distribution and concurred the main spatio-temporal patterns described by the authors within the article. 

Reviewer #2 – 5°. The authors note that per the data in Table 1, the intermediate (5-65 year) and older age groups were more likely to be female, however, Figure 2 appears to show that for all age groups other than those aged 24-30, 30-35 and 85-90, cases were more likely to be male. Suggest clarifying why age was collapsed into such broad groupings (particularly the intermediate category) as potentially significant differences between age groups (as shown in Figure 1) seem to have been lost through aggregation of cases in the intermediate age group, potentially changing the direction of association. From the data presented in Table 2, it appears that the data for the intermediate age group could have been separated into three or four different categories/groupings to create similar sized age groupings for analyses. Assessment of age via smaller groupings may have produced different findings, more similar to Figure 2. 

Response Reviewer #2 – 5°. The authors agree and absolutely concede that some data were lost via collapsing (discretization) into one relatively broad category, however, SatScan analyses can take days and weeks to undertake when scan tuning is included, and it was thought best to focus on the most vulnerable subgroups, as delineated in the literature. Previous epidemiological research has and continues to use these groupings in order to 1. focus on the most susceptible in society, and 2. Ensure overall study efficacy i.e., if much finer categories are employed, datapoints/sample numbers are dramatically decreased, and no clusters will be identified (it is very much a case of balancing level of detail with level of interpretability!). A similar approach has previously been used, validated and published by Cleary et al (2021) and Boudou et al (2012).

Reviewer #2 – 6°. For those cases that were not able to be geolinked, were these randomly distributed or did this result in the complete loss of certain areas for spatial analyses? 

Response Reviewer #2 – 6°. This is a good question. We can confirm that cases that were not successfully geo-linked were randomly distributed. As such, paediatric cases represented 22.7% of total cases in overall cases number and 22.8% of total non geographically linked cases. Similar observations were made for other age groups and gender groups. 

Reviewer #2 – 7°. It would be helpful for the international reader if the authors could clarify what months Irish summer occurs in. Winter is noted as being Dec-Feb, however the authors refer to May as being during summer – in other countries in the northern hemisphere May is considered to be in spring (typically March-May). Similarly, Figure 4 appears to show that May is a summer month and that February occurs during Spring (not winter as stated in the results section). Suggest using consistent categorizations for season throughout the manuscript. 

Response Reviewer #2 – 7°. Irish seasons are indeed classified using a different approach to that employed in other regions in the Northern Hemisphere, inherited from the Celtic Calendar. Mistakes relating to the months cited above have been corrected/amended throughout the manuscript. Similarly, additional information describing was added within the method section (2.2). 

“Irish seasonal calendar, traditionally used in the ROI and different from other European countries, was employed for describing campylobacteriosis seasonal patterns (i.e., Winter: November to January, Spring: February to April, Summer: May to July, Autumn: August to October).”

Reviewer #2 – 8°. In Figure 4, suggest clarifying the case numbers in the legend as (n=3,015) etc. instead of (3,015n) so this is clearer for the reader. 

Response Reviewer #2 – 8°. Figure 4 has been amended as requested. We agree this improves overall clarity and replicability for the reader 

Reviewer #2 – 9°. Suggest clarifying in the results section which method produced each set of results. i.e., were the results for seasonal clustering in section 3.3. obtained through space-time scans? 

Response Reviewer #2 – 9°. Thank you. Perhaps we were not clear. Results of seasonal clustering were not obtained through space-time scanning: space-time scanning is not undertaken to decompose or delineate raw time-series data into seasonal variation patterns, this is only achievable via seasonal decomposition (and specifically the LOESS approach used in the manuscript). Seasonal variations on a county-by-county basis were employed as input variables, with hierarchical cluster analyses subsequently used to identify county clusters (i.e, an ensemble approach).

Details related to this aspect/approach are now presented in Section 2.2.: 

“For the current study, seasonal signals obtained from individual (i.e., county level) seasonal decompositions were employed as inputs for cluster analysis to identify counties exhibiting similar temporal patterns, using a hierarchical clustering approach via the Ward2 algorithm.”

Reviewer #2 – 10°. Were the results of each methodology similar where applicable or did any methods produce conflicting findings?

Thank you. As presented, results emanating from each statistical methodology were used to identify and highlight the spatial and temporal patterns of campylobacteriosis in the Republic of Ireland during the study period, objectively compared, and discussed . While some relatively minor variations were observed as might be expected to occur via differing mathematical and statistical algorithms, no significantly or overarchingly contradictory findings were noted by the authors.

Reviewer #2 – 11°. In the results section the authors note that a positive association was found between female cases and urban settlements, and male cases were more likely to occur in commuter regions and rural areas. Suggest providing additional discussion re: why this might be. I.e. are female cases potentially more likely to be food-related and male cases more likely to be animal related if males are more likely to work in agricultural occupations? 

Response Reviewer #2 – 11°. The authors would like to sincerely thank the review for their insightful comments. We agree that some further discussion was warranted with respect to these points. Accordingly, some additional information were added to the manuscript in relation to that aspect (see below), however it is also important to point out that these hypotheses which likely accurate cannot be conclusively proven via a national scale ecological study as presented here. A matched case/control study would be required in this case: 

“Similarly, as shown (Table 2), higher likelihoods of contracting campylobacteriosis were noted amongst males residing in rural (aOR: 1.005, CI 95% 0.92 - 1.09) and commuter (aOR: 1.173, CI 95% 1.07 -1.29) areas. This study finding may reflect a higher risk of transmission via direct animal contact among the rural and male sub-population. While occupational status was not highlighted as a risk factor for campylobacteriosis in Ireland by Danis et al. (2019), Green et al. (2006) found that the male population under 19 years of age in rural areas of Manitoba (Canada) and engaging in agricultural practices, represented the highest risk group for campylobacteriosis. “

Similarly, food consumption patterns among commuters may differ from non-commuters – e.g., increased tendency to purchase fast food. As the authors noted, FSA Ireland and other studies have previously found consumption of RTE food to be a risk factor for campylobacteriosis. “

Dear Reviewer, additional information related to chicken consumption were added within the discussion. 

“Similarly, Consumption of undercooked chicken and contaminated ready-to-eat (RTE) foods during the summer months have been identified as important risk factors for sporadic Campylobacter spp. infection in both the Republic of Ireland and Northern Ireland (Danis et al. 2009).”

Reviewer #2 – 12°. As above, it would be helpful to provide some additional discussion re: potential agricultural explanations for identified hot and cold spots. For the reader who is unfamiliar with the ROI, it would also be interesting to provide some discussion re: how the location of hot and cold spots relates to the distribution of reservoir agricultural species (e.g., cattle, small ruminants), contact with which is a known risk factor for infection. Previous studies have found that prolonged ongoing exposure can induce immunity to infection, which could be a potential explanation for the

identification of cold spots in rural/agricultural areas (if applicable), particularly in older age groups. 

Response Reviewer #2 – 12°. Dear reviewer, the authors agreed that identified spatial hotspots and cold spots are likely related to agricultural practices for rural areas. This point was stated in the discussion and further details were added.

“Accordingly, the southern part of the island is the primary location of intensive dairy farming in the ROI[42], while the Northern part of the country is characterised by mixed agricultural activities such as small dairy farming, arable farming, or cattle and sheep grazing [43].”

“In contrast, infection cold spots (Figure 7) and low cluster frequencies (Figure 8) were localised in Western (i.e., Kerry, Mayo) and Northern areas (i.e., Donegal, Sligo, Leitrim, Louth) potentially reflecting local agricultural practices with a lower concentration in livestock and dairy farming industry [43] and local geography, characterised by a general higher annual precipitations [48].”

 Additional analyses are currently ongoing, using an ensemble of statistical modelling (i.e., GLM, Gradient Boosting, RF…) to address specifically the spatial risk factors of respective hotspot and cold identified by this study. 

Reviewer #2 – 13°. The authors note that high rates in children may be due to increased susceptibility, while older age groups may be more likely to be symptomatic – this appears to indicate that observed incidence in the <5 age group may be an underestimate if children are more likely than older age groups to be asymptomatic or to only have mild illness (and less likely to be tested). 

Response Reviewer #2 – 13°. The authors would like to sincerely thank the reviewer for their attention to detail – this was a typo: the authors of course meant to report that the infection within older age groups tend to become more “asymptomatic” than for the paediatric group (who are also more regularly tested due to likely increased severity, guardianship concerns, etc). This has now been amended within the manuscript, as follows.

“The high incidence of campylobacteriosis among children under 5 years old is likely explained by a lower immunological profile within this age group, with the infection tending to become increasingly asymptomatic among higher age groups [23]. A study by Havelaar et al. [24] suggests that children and individuals having previously experienced a primary exposure to the pathogen are more likely to develop antibodies and therefore less likely to develop symptoms during a secondary exposure.”

Reviewer #2 – 14°. As a note: suggest defining ROI at first use and using this consistently throughout the manuscript 

Response Reviewer #2 – 14°. Thank you. The abbreviation for Republic of Ireland (ROI) was has been added upon first use of the term (Introduction section) and is not used consistently throughout the manuscript.

---

## [Decision Letter · Decision Letter 1]

6 Sep 2023

An ecological study of spatiotemporal dynamics and drivers of domestically acquired Campylobacteriosis in Ireland, 2011–2018

PONE-D-23-16493R1

Dear Dr. Boudou,

We’re pleased to inform you that your manuscript has been judged scientifically suitable for publication and will be formally accepted for publication once it meets all outstanding technical requirements.

Kind regards,

Csaba Varga, DVM MSc PhD

Academic Editor

PLOS ONE

Additional Editor Comments (optional):

Reviewers' comments:

Reviewer's Responses to Questions

**Comments to the Author**

1. If the authors have adequately addressed your comments raised in a previous round of review and you feel that this manuscript is now acceptable for publication, you may indicate that here to bypass the “Comments to the Author” section, enter your conflict of interest statement in the “Confidential to Editor” section, and submit your "Accept" recommendation.

Reviewer #1: All comments have been addressed

Reviewer #2: All comments have been addressed

2. Is the manuscript technically sound, and do the data support the conclusions?

Reviewer #1: Yes

Reviewer #2: Yes

3. Has the statistical analysis been performed appropriately and rigorously? 

Reviewer #1: Yes

Reviewer #2: Yes

4. Have the authors made all data underlying the findings in their manuscript fully available?

Reviewer #1: No

Reviewer #2: No

5. Is the manuscript presented in an intelligible fashion and written in standard English?

Reviewer #1: Yes

Reviewer #2: Yes

6. Review Comments to the Author

Reviewer #1: I commend the authors for their very careful and thorough review and revision. I agree with the statement that this is a foundational paper and not adjusting for some covariates in the cluster detection approach is acceptable. I believe the paper is now ready for publication consideration. Well done.

Reviewer #2: Thank you for the opportunity to review a revision of this manuscript. The authors have appropriately responded to/addressed my previous comments and I have nothing further to add. Thank you!

7. PLOS authors have the option to publish the peer review history of their article (what does this mean?). If published, this will include your full peer review and any attached files.

Reviewer #1: No

Reviewer #2: No

---

## [Editor Report · Acceptance letter]

12 Sep 2023

PONE-D-23-16493R1 

An ecological study of the spatiotemporal dynamics and drivers of domestically acquired campylobacteriosis in Ireland, 2011–2018 

Dear Dr. Boudou:

I'm pleased to inform you that your manuscript has been deemed suitable for publication in PLOS ONE. Congratulations! Your manuscript is now with our production department. 

Kind regards, 

on behalf of

Dr. Csaba Varga 

Academic Editor

PLOS ONE